# Customizable FPGA-Based Hardware Accelerator for Standard Convolution Processes Empowered with Quantization Applied to LiDAR Data

**DOI:** 10.3390/s22062184

**Published:** 2022-03-11

**Authors:** João Silva, Pedro Pereira, Rui Machado, Rafael Névoa, Pedro Melo-Pinto, Duarte Fernandes

**Affiliations:** 1Algoritmi Centre, University of Minho, 4800-058 Guimaraes, Portugal; a82040@alunos.uminho.pt (J.S.); a81756@alunos.uminho.pt (P.P.); rafael.accn@gmail.com (R.N.); pmelo@utad.pt (P.M.-P.); duarte.fernandes@dtx-colab.pt (D.F.); 2Associação Laboratório Colaborativo em Transformação Digital—DTx Colab, 4800-058 Guimaraes, Portugal; 3Bosch Company, 4700-113 Braga, Portugal; 4Centre for Research and Technology of Agro-Environmental and Biological Sciences, University of Trás-os-Montes and Alto Douro, 5000-801 Vila Real, Portugal

**Keywords:** convolutional neural network (CNN), hardware accelerator, field-programmable gate array (FPGA), light detection and ranging (LiDAR), quantization, object detection

## Abstract

In recent years there has been an increase in the number of research and developments in deep learning solutions for object detection applied to driverless vehicles. This application benefited from the growing trend felt in innovative perception solutions, such as LiDAR sensors. Currently, this is the preferred device to accomplish those tasks in autonomous vehicles. There is a broad variety of research works on models based on point clouds, standing out for being efficient and robust in their intended tasks, but they are also characterized by requiring point cloud processing times greater than the minimum required, given the risky nature of the application. This research work aims to provide a design and implementation of a hardware IP optimized for computing convolutions, rectified linear unit (ReLU), padding, and max pooling. This engine was designed to enable the configuration of features such as varying the size of the feature map, filter size, stride, number of inputs, number of filters, and the number of hardware resources required for a specific convolution. Performance results show that by resorting to parallelism and quantization approach, the proposed solution could reduce the amount of logical FPGA resources by 40 to 50%, enhancing the processing time by 50% while maintaining the deep learning operation accuracy.

## 1. Introduction

The increased focus on research and development on intelligent systems has been growing with different technologies providing different applications on a variety of complex systems. Concerning autonomous vehicles, the main motivation centers on reducing human interference while driving, thereby reducing the likelihood of road accidents caused by human error, improving road safety [1,2]. With that, a highly detailed perception of objects surrounding every vehicle is required, enriching the perception capabilities of these vehicles, allowing thus an efficient capture of information about the localization, classification, and tracking of such vehicles.

In this scope, LiDAR sensors have been highlighted as a technology that allows a description of the vehicle surrounding by means of point cloud data, being exploited in the literature as an augmentation to RGB cameras as a standalone solutions [3,4,5]. Currently, deep learning models are widely used to process point cloud data, provided from LiDAR sensors, to extract relevant information that may be used in a mechanism for object detection and localization [6,7,8,9,10,11,12]. Different studies suggest a higher precision rate during object detection and classification using deep learning models aside from the classical point cloud algorithms that manually extract features using machine learning techniques [3,6]. However, these models have some drawbacks regarding real-time execution (being often incapable of providing inference time lower than the sampling rate of LiDAR sensors, 10/20 Hz) and the required resources for computation.

Deep learning models take advantage of recent developments of convolution neural network architectures, resulting in pipelined architectures with different configurations that may vary regarding the number of convolution blocks, its parameters (stride size, kernel, pooling operation), presence/absence of activation functions, and normalization methods through each pipeline structure. Most of those systems are deployed in graphics processing units. Besides providing higher performance, it is necessary to deploy those systems in edge devices, limited by tight timing constraints, which usually are not considered on GPUs or central processing units implementation approaches [13,14,15,16,17].

This work is not intended to create a fully convolutional neural network on an FPGA, instead, the main goals center on the implementation of a configurable convolution module, evaluation of the impact on performance by applying optimization methods such as quantization and parameter sharing, and integration of the convolution module on different CNN architectures. To this end, parallelism approaches for the processing element (PE) were studied and implemented and performance regarding PE gains was compared. Our convolution module which implements a PE based on work [18] provides an improvement of that work by increasing inside the same module the ability to compute rectified linear unit (ReLU), padding, and max pooling. On top of that, the convolution module can be configured to match the features of a convolution block addressed in the literature as part of an object detection model. It means that varying the size of the feature map, filter size, stride, number of inputs, number of filters, and the number of hardware resources required for a specific convolution it is only necessary to change its parameters at the instantiating time. Furthermore, the quantization technique’s impact on the performance of a 3D object detection model, regarding metrics accuracy and inference time, was studied and implemented. Both developments were correctly validated in different applications, where we verified the correct operation and process efficiency on both image and point cloud processing. To the best of our knowledge, this is the first work studying the quantization influence on model performance as a function of the model depth.

The paper is organized as follows: Section 2 describes some applied techniques on 3D object detectors based on point clouds data. The section ends with a description of different techniques and optimization methods applied to CNN architectures are also discussed. Section 3 describes the proposed architecture and filter iteration through each instantiated convolution block. Section 4 describes the most relevant parts regarding hardware implementation, block interface, and interactions. Section 5 presents obtained results for three different studies: (1) generic image convolution with filter applications changing the values of its parameters regarding convolution block, presenting an evaluation between processing time, level of parallelism, and consequently resources usage; (2) study about quantization influence on a simple CNN architecture, using the MNIST dataset; (3) the last validation promotes a replacement of a software convolution layer of the PointPillars model running on a laptop with values obtained from the hardware processing using the implemented IP. Finally, Section 6 provides thesis conclusion as well as some considerations for future development.

## 2. State of The Art

This section provides a brief description of related works and improvements on hardware accelerators for convolutional blocks. Convolution neural network (CNN) architectures are widely used in image recognition and are efficient for object detection, localization, and classification [19]. There are several key CNN architectures, namely LeNET [20], VGGNet [21], and ResNet [22], generally speaking, they are typically built using fundamental layers like convolution, pooling, and fully connected. Convolution layers perform a vital role in how CNN architecture operates, being responsible for around 90% of all computation. Thus, this section focuses on presenting a brief description of 3D point cloud model architectures and developments of hardware accelerators already implemented.

### 2.1. Deep Learning for 3D Point Cloud

Recent works on 3D point cloud models present a sequential architecture that is split into three stages: (1) data representation, (2) feature extraction, and (3) detection modules. Stage (1) processes the data from the LiDAR sensor and organizes it as a structure that can be easily readable and processed by the following stage. Concerning literature, those structures are created as “Voxels”, “Frustums”, “Pillars”, or 2D projections [6,7,9,10]. Stage (2) presents the feature extraction process for a given point cloud. The last stage (3) is defined by its output values which lead to possible object detection. Those outputs describe the probability of object classification, bounding box regression, and object orientation.

These models have in common the conversion of the input into a pseudo-image upon the first stage, (1) data representation, which means that 2D representations are applied in further convolution layers. The convolution operation is a fundamental process for feature extraction, providing object classification and bounding box regression.

### 2.2. Convolution Implementations in FPGAs

Convolutional layers present in CNN architectures introduce high computational costs, due to the extensive number of arithmetic operations, parameter sharing, and memory access. These issues not only increase the amount of hardware resources required but also hampers some complex CNN from achieving their full potential as they are not able to output inferences in a real-time manner. Therefore, migrating convolutional blocks to hardware aims at mitigating those problems, providing a hardware architecture optimized for these operations and, consequently, more reliable, efficient, and time-consuming with fewer resources. Research works [14,18,23] provide advanced architectures that take advantage of parallel computation.

#### 2.2.1. Sliding Window Dataflow

In the proposed architecture in work [23], the processing element unit has a MAC unit and multiply and accumulate operation. Besides the MAC unit, PE blocks hold on-chip memories for input data, weight values, and outputs results. The proposed architecture in work [14] presents parallel multipliers to compute all output products in a single clock cycle, and an adder tree to aggregate all outcomes. The adder tree is predefined to hold a fixed number of multiply operations, which means is limited to a certain convolution process regarding filter size. Both works [14,23] provide hardware blocks optimized for processing time. However, these solutions rely on redundant on-chip memory access, promoting high energy consumption.

#### 2.2.2. Rescheduled Dataflow Optimized for Energy Efficiency

In work [18], a processing element is featured with a multiply and an accumulate operation, MAC unit. Input feature map data and weight data needed for the convolution process are loaded from off-chip memory to on-chip memory. Once on-chip memory is connected to the processing units, memory access requires low energy consumption as data transfer is faster than between off-chip and on-chip memories. Data present on on-chip memory are never discarded, providing access reduction to off-chip memory, decreasing system latency. After finishing the convolution process, the output values are sent to an on-chip memory reserved to hold output values. If necessary, output data stored on output on-chip memory is transferred to off-chip memory for data analysis.

### 2.3. Optimization Methods

Deep learning algorithms are usually implemented in software using 32 bit floating-point values (FP32). Migrating a deep learning algorithm to an ASIC or an FPGA requires for a bit width reduction which is possible using the quantization technique [24,25,26]. A quantized model and a non-quantized model execute the same operations, however, a quantized model with bit-width reduction promotes a memory reduction and allows the execution of more operations per cycle. This memory reduction allows a more compact model representation, which leads to a better deployment in a hardware platform. For hardware, implementation is intended to convert a 32 bit floating-point value to a 16/8/4 bit fixed-point value INT (fixed-point expression), respectively [27]. The bit reduction may lead to a considerable accuracy gap on full precision models as suggested by [28].

Therefore, it is necessary to achieve a trade-off regarding model accuracy, model parameters, and hardware (HW) performance. The work in [29,30] presents a method that takes full advantage of a DSP block for 8 bit quantization. However, the trade-off between accuracy and inference time might be required and applied whenever possible, therefore, this study provides insights about the model degradation for various model depths, i.e., number of layers.

## 3. Convolution Hardware-Based Block

The proposed block IP was designed taking into consideration the developments in deep learning models for object detection addressed in the last five years. Our work based on rescheduled dataflow, exploited by work [18], provides a different implementation since we create an IP capable of computing not only convolution but also rectified linear units, padding, and max pooling, that can be configured by simply changing its parameters and adapting it to different CNN architectures. Besides that, quantization was applied to each weight value which leads to a reduction in bit-width leading to parameter sharing and promoting a DSP resource usage decrease. Therefore, the parameters and the range of values that might change from model to model were identified, in order to tailor the block for any desired architecture whenever required.

### 3.1. Block Architecture

As Figure 1 depicts, the architecture of the convolution block proposed in this work comprises the following three distinct modules: processing element, rectified linear unit, and max pooling. The convolution block has six parameters that provide different architecture configurations. Five of the six parameters are related to the theoretical convolution process. The ability to change the six parameters provides an advantage to other works since different configurations are possible. We defined the number and set of configurable parameters according to the convolutions layers implemented in the literature for 3D object detection. These parameters are as follows: (1) feature map size, (2) weight/filter/kernel size, (3) stride value, (4) padding value, (5) maxpool enable/disable, and (6) number of DSP block per convolution block.

Inside the convolution block, on Figure 1, processing data flows through modules sequentially, i.e., PE result output is forwarded towards the following module, ReLU, which after performing its operations, forwarded its output to the next module, maxpool, whenever parameter (5) is enabled. A “controller” module ensures precise BRAM addresses management, providing an ordered data transfer from a block RAM to a convolution block.

### 3.2. Processing Element

The processing element module is considered a low-level block inside the proposed architecture. All convolution processes, meaning multiply and accumulate operations, are carried out by the PE module. At the same clock instant, three new values are fed to the input ports of each PE module, as Figure 2 illustrates: (1) feature map value, (2) filter (weight) value, and (3) previous PE output value.

We explore the DSP block usage to reduce the amount of resources required, as DSP templates provide a favorable trade-off between developed configuration and resources usage. Flexibility during architecture implementation is ensured as every DSP signal and parameter is changeable during instantiation, otherwise using another of the three types of inference leads to lower flexibility and higher resource consumption.

### 3.3. Memory Access and Dataflow

In order to reduce the limitations of the sliding window approach discussed in state of the art, namely on memory access and redundant data, our solution proposes a distinct approach of the research works [14,18], being thus inspired by the research work [23]. The proposed architecture follows sequential processing data, meaning that each FM value is fed one by one to the processing module. This mechanism ensures that the system architecture only changes when applying filters with different dimensions, providing a scalable and stable architecture.

The dependency data flow chart in Figure 3 illustrates graphically how each outcome from a multiply operation must be connected to other multiply operations, where: (1) *F*_xx refers to input FM data; (2) *W*_xx refers to weight/filter/kernel data; (3) *O*_xx refers to output FM data; (4) for each vertical line, with 9 values corresponding filter size, all data are computed simultaneously; (5) each vertical line is processed with one delay clock cycle; (6) black arrows refer to input data of the next processing element, added up with the corresponding multiply outcome.

This architecture provides a modular configuration, meaning that changing the input FM dimensions or filter dimensions leads to easy reconfiguration without losing architecture integrity and maintaining the correct convolution function. Equation (Equation 1) presents the required processing time as a function of the characteristics of the input data, namely FM size (*FM*), number of input FM (*num.FM*); level of parallelism required (*Num.PEs*), and board clock (*ClockFreq.*).
(1)$$\frac{\frac{FM^{2}}{Num.PEs}\times Num.It\times Num.FM}{ClockFreq.(\mathrm{MHz}))}\left(\mu s\right)$$

As Figure 4 depicts, for a filter of 3 × 3 the processing element module is built with nine DSP blocks. As illustrated, in every N DSP block, a ShiftRAM is placed, N being equal to filter size (3 for the above example). Each ShiftRAM introduces a predetermined delay which serves as synchronization for data processing in the next DSP block. The predetermined delay depends on feature map and filter dimensions. The delay value is obtained by subtracting the FM dimension from the filter dimension. At the end of this section, a different configuration is present regarding quantization models, for that, DSP block and ShiftRAMs usage decrease due to bit width reduction making it possible to have two multiplication and one accumulate operation for a single DSP block.

### 3.4. Board Resources-Driven Architecture

To provide a convolution block that can be tailored for a certain application, three new configurable parameters were added which will influence the number of PE modules for simultaneously processing. These parameters are as follows: (1) DSP available, (2) BRAM available, and (3) BRAM memory bit. The parameters (2) and (3) emerge from memory limitation, allowing the user to specify the amount of memory available for a set of convolution blocks.

Considering hardware resource limitations as a constraint of many CNN hardware implementations, parameter (1) DSP availability is added to indicate the maximum number of DSP blocks that can be distributed through the PEs components of our convolution block. In order for our block to automatically adapt its architecture to the resources available on the target board, parameters (2) and (3) are considered for specifying the amount of filters that can be simultaneously processed. It will ensure that the number of convolution blocks is limited by the memory available. The minimum memory required for one filter application is given by output FM dimension and output data bit width. Equations (Equation 2) and (Equation 3) provide information on the number of parallel filters and number of PEs modules that can be instantiated:(2)$$NumberofParallelfilter=\frac{BRAMavailable\times BRAMmem.bit}{OUTFMsize\times OUTFMsize\times OUTwidth}$$
(3)$$NumberofPEblocks=\frac{Numberoffilterstoapply}{Num.ParallelFilters\times KERNELsize\times KERNELsize}$$

### 3.5. Filters Iteration—Control Module

Each convolution block only has access to the values corresponding to a single filter, meaning that for the proposed convolution block architecture, the number of blocks required to instantiate and to assure the correct operation in parallel matches the number of filters desired to apply simultaneously. As illustrated in Figure 5, all convolution blocks are instantiated inside a layer block, which has a state machine that controls each stage for data processing. Each stage has distinct functionalities since reading memory from input BRAMs, data transfer to lower levels modules, and write memory to output BRAMs.

As Figure 5 presents, in idle stage (1), a reset is performed to all internal registers ensuring transition for the second stage, providing weights load from BRAM to PE modules. State machine continues in load weights stage (2) until all weight values are transferred to PE modules. Data transfer ends after Kernel_size*Kernel_size clock cycles. In load in/process FM stage (3), the convolution process begins, all FM data are transferred and processed one by one, as mentioned at the beginning of the design section. The final stage (4), indicates that a new FM was generated being the correspondent output data in memory.

### 3.6. Optimization Methods

This section presents two methods for optimizing the convolution operations on the proposed convolution block architecture by including features such as parameter sharing and quantization. The quantization process is considered an efficient and fundamental approach for compression of such model targeting resource-constraint devices, such as edge devices. Due to input feature map and weight bit width reduction performed by quantization technique from 32 bit floating point to 8 bit fixed point, the previously described architecture should be updated to handle the new data format.

#### Architecture Reconfiguration with 8 Bit Quantization

In research work [29], two parallel MAC operations compute two dot products. For implementing these two MAC operations, it is necessary to use input port D from the DSP48Ex block. For the two multiply operations, it is necessary to apply the following equation P = (AB + DB) + C. The value that port A receives is arithmetically left-shifted by 18 bits. Data in the D register are stored in the least significant bits positions and data in the A register are left-shifted 18 bits to ensure that the outcome from the pre-adder module does not lose any weight value for each computation.

As Figure 6 illustrates, the reconfigured architecture has, for a single DSP block, two MAC modules, promoting fewer shift RAM blocks usage since more DSP blocks are directly connected. As Figure 6 presents, four DSP blocks have a direct connection which refers to the filter/kernel dimension (in this case a filter of 4 × 4 is applied). The proposed architecture needs to ensure that outcomes from MAC modules are correctly sent for the following DSP blocks. In case of a direct connection, each result is directly sent to the C input port. This new DSP block receives as input the first weight value of the following four consecutive Weight values. After finishing processing the first four weight values, represented in blue, results from the last MAC module are sent to the first shift RAM. Output data from a shift RAM has the same delay as presented for the first proposed architecture, which varies regarding weight/filter/kernel dimension, FM dimension, and stride value. The shift RAM block receives one value that holds two different results regarding blue and red DSP block computation. For a correct accumulation of the first red DSP block, output data from shift RAM need to be 8 bit right-shifted.

## 4. Implementation

All code was implemented using Verilog with all the modules previously described being implemented on the programmable logic (PL). To validate the implemented hardware IP, after test bench validation for behavioral and timing simulations, it is necessary to validate the IP on an FPGA board. The tests present in the results section were deployed on a Zybo Z7:Zynq-7000 board [32]. To evaluate the consumed resources by the proposed IP an input image of 252 × 252 dimension was used, which is a typical size widely adopted in deep learning models. To build an entire CNN architecture with our convolution IP, it is only required to instantiate the correct number of convolution blocks while ensuring that data flow is correctly performed.

Using our IP interface, the user is able to configure at design time each parameter addressed to a convolution. This means that for the application of the module in a real CNN deployment it is mandatory to determine the correct number of convolution layers and each parameter value, before implementing it on an FPGA. As found in the literature, a major problem regarding CNN implementation in hardware centers on resources limitations, such as DSP blocks and memory. Considering that constraint, three parameters (DSP available, BRAM available, and BRAM memory bit), which are used to define the number of filters processed at the same time, were added.

Figure 7 and Table 1 presents the resources required for implementing a convolution of a 3 × 3 filter to an image of 252 × 252 using only one PE block (9 DSP blocks) with a clock source of 100 MHz. The figure shows the resources required for IP (design_1_layer_2_0_0_synth_1) at the bottom. The total of resources required for managing the ARM processor, DMA module, blocks of RAM, and our IP is displayed as impl_1. Additionally in the figure are illustrated how resources are split on the FPGA device. The resources related to the implemented IP are highlighted with red, inside the purple square box. The other resources are related to AXI modules, DMA controller, BRAMs, and processing system.

## 5. Results

The tests presented in this section were obtained with each image loaded to an SD card that is connected to the Zybo board. Using the ARM processor, provided by the Zybo Z7 board [32], the DMA controller was configured to store input image data in DDR memory. Thus, before the convolution process begins, the DMA controller sends input images to the RAM blocks that are directly connected to the proposed IP.

### 5.1. Generic Convolution

To evaluate the correct functionality of the convolution block, it was applied different filters to a set of figures, changing the values of the stride, padding, and maxpool parameters. The dimensions of input FM are 252 × 252, with a filter application of 3 × 3. Figure 8 depicts each output from applying the same 3 × 3 sharpen filter using a stride of 1 and 3, resulting in outputs of 250 × 250 and 84 × 84, respectively.

#### 5.1.1. Parallelism Influence on Processing Time

Considering that in the previous example we used a 252 × 252 image with a filter application of 3 × 3, the study begins with the simple case which uses only one PE block, providing less resource usage and higher processing time. Each test iteration increases the number of PE blocks to a maximum of 100 PEs. For a filter of 3 × 3 and applying up to 100 PEs blocks, 900 DSP blocks are required. As illustrated by Figure 9, the instantiation of only one PE module leads to a higher processing time. The application of two PE modules in parallel leads to a reduction in 50% of the previous processing time. Applying three PE modules in parallel reduces the processing time by 65%. As illustrated, after 20/30 PE modules in parallel the processing time reduction is less relevant.

The blue curve in Figure 9, represents theoretical processing time values previously computed regarding PEs block usage, the orange curve represents processing time values during hardware IP processing measured in the same previous conditions. A slight deviation in theoretical values and hardware measures appears on the graph since for “theoretical time” we do not consider the initial clock cycles that do not generate valid outcomes, as explained in the design section.

#### 5.1.2. Block RAM Influence on PEs/Number of Parallel Filter/Processing Time

For the following example, the IP parameters were configured as: kernel size = 3, FM size = 252, padding = 0, stride = 1, maxpool = 0, input FM channels = 64, num of filters = 32, DSP block available = 1000, and clock source of 100 MHz.

As Figure 10 presents, when only one block of RAM is used only one filter is applied (orange line), consequently, the maximum possible number of PE blocks are used. Increasing the number of blocks of RAM leads to an increase in the number of parallel filters. Consequently, the number of PE blocks per convolution block decreases since more PE blocks are spread to different filters reducing the number of available PEs for a single convolution block. Increasing the number of parallel filters provides a decrease in processing time, as illustrated in the graph on the right side. However, after the two lines intersect in the upper graph, processing time increases. This can be explained once more filters are computed simultaneously which results in fewer DSP blocks allocated per convolution block. Thus, once fewer DSP blocks are used in a single convolution block the processing time increases.

### 5.2. Quantization Influence Study

This section presents a study regarding the quantization impact on two different deep learning models based on a CNN architecture. The MNIST model is used for handwritten digit detection in an image. The other model refers to a 3D object detector and classifier model based on point clouds, namely, PointPillars. The two deep learning models supported by the software version use the input data and weight values in a 32 bit floating-point format. This study intends to convert the aforementioned data from single-precision to an 8 bit fixed-point format.

#### 5.2.1. MNIST Dataset Model

The model configuration used for validation is built with two convolutions layers, fully connected and softmax. Figure 11 illustrates what changes were performed to validate the IP functionality. In the first study, the values were hardware computed using the developed IP. In a second study, with an integration of eight new convolution layers, the quantization methods through all layers were implemented. As represented inside the red square, the input values of the first convolution layer are sent to hardware IP. The output values from hardware processing, which are quantized due to hardware conversion from 32 bit floating-point format to 8 bit fixed-point format return to the software (SW) model to feed the next convolution layer, the input values of the second layer are replaced with hardware values.

The fixed-point representation is expressed as Qi.d, where i refers to the integer part of the fixed point, in other words, the number of bits present on the left side of the fixed point, and d refers to the number of bits on the right side of the fixed point, as a decimal fraction. The weights values for the first convolution layer were quantized with three different quantization levels: Q2_6, Q3_5, and Q4_4. For each one of them, the output FM quantization level was tested with different configurations providing a better study on classification score due to weight and output FM value quantization.

##### Convolution Weights and Bias Quantized

In a second iteration, illustrated in Figure 11 by the green box, the study was extended to apply quantization to all layers of the MNIST model. To the previous MNIST model a few more convolution layers were added, with that we intend to verify the quantization impact on deeper CNN architectures, so we added eight (8) new convolution layers. The quantization was performed using the same IP that was used in the previous study. With this, we can obtain a quantization result that is approximately close to a possible implementation of the entire architecture in HW. This means if we intend to implement the entire model on hardware, which is not the goal with this study, the results expected will be close enough to the results presented in Figure 12.

For this study, only Q1_7 and Q2_6 configurations were applied, another type of configuration such as Q3_5 or Q4_4 leads to worse results since the number of fractional bits is reduced, leading to accuracy loss. The previous study only uses as input one image, this study applies four different images with handwritten digits of 1, 2, 3, and 4. The graphs depicted in Figure 13 and Figure 14 present the obtained score for the correct classification. In other words, associates each of the four images with a correct classification (image 1—digit 1, image 2—digit 2, and so on). The blue bar on graphs represents the score values obtained using an SW-only version. The gray bars, from left to right, represent the score obtained by applying quantization to all layers of the MNIST model and quantization only on convolution layers, respectively. As expected, applying quantization to all layers leads to a slight score reduction. This reduction, presenting a higher error on image 2 with 1.88%, varies depending on which image is used for classification. Nonetheless, it is almost insignificant, leading to a robust classification for each of the four different images.

#### 5.2.2. PointPillars Model

##### Quantized Convolution Weights

On the PointPillars model, we verified how quantized convolution weights of backbone convolution layers affect score detection and interest over union (IoU). For this analysis, we adopt the performance metric mean average precision (mAP) while using the Kitti dataset. The mAP is used to measure the accuracy of an object detector regarding its precision and recall. Using a Python script, it was possible to obtain a result checkpoint using different frames of the Kitti dataset, the result checkpoint provides mAP values for the evaluation scenarios of bounding box (BBOX), bird’s eye view (BEV), 3D, and average orientation similarity (AOS).

Figure 15 presents the mAP metric value for each of the four scenarios regarding the three difficulty levels, using SW-only version with 32-bit weight values and quantized version with 8-bit. In Figure 15, below each scenario presents three bars regarding the three difficulty levels (from left to right refers to easy, moderate, and hard). Each difficulty level value for the SW-only version is directly compared with the correspondent value for a quantized version, which means, for the BBOX metric, it is possible to evaluate the SW and Q_CONV result value that combines the same bar color. In this case, the SW-only version for the easy level results in an mAP of 83.73% while the quantized version results in 80.84%. The same evaluations are applied for the remaining metrics. This mAP degradation can be explained by the precision loss of the feature maps, but also due to the fact that the outputted classification scores are affected as will be further shown, suggesting that the score threshold should be adjusted to reduce the metric performance loss.

Different frames were evaluated through Kitti Viewer, thus classification score and IoU values were collected using checkpoint files resulting from the previous step, i.e., evaluate using the SW-only version model and quantized model version (quantized weight values from the backbone convolution layer). After analyzing every frame regarding classification score, IoU, and object distance to LiDAR sensor, the graphs in Figure 16 were plotted. Each blue dot represents a detected object for a given frame, describing object depth regarding the LiDAR sensor and its correspondent classification score value and IoU value. This study tries to establish a direct relationship between the obtained classification score value for a given object and its distance to the LiDAR sensor.

As showcased in the graphs of Figure 16, greater object accumulation can be indicated for classification scores higher than 0.6. These objects are located at a maximum distance of 30 m from the LiDAR sensor, representing 65.5% of objects for the SW-only version using 32 bit weights. For the quantized version, using weights with 8 bits, object accumulation continues high under 30 m distance from the sensor, with higher score values (total of 65.98% of objects). As for the IoU metric, the graph in Figure 16 presents a higher object concentration with higher IoU values for distances up to 30 m, resulting in 62.02% of the total objects. For the quantized model, regarding IoU values, the graph presents a higher scatter of the object’s distribution, due to bit reduction, which leads to some null values, resulting in a quantitative reduction to 51.55%.

The two graphs in Figure 17 illustrate how the score and IoU error is spread regarding object distance from the LiDAR sensor. In each graph, it is possible to verify that some error values are negative, meaning that the classification score or IoU value is higher for the quantized weights model version, providing an improvement of detection accuracy once score and IoU metrics increase. The graph that represents the error score shows a higher accumulation of values for an error under 0.05 and over −0.05, representing a point accumulation of 61.63% for all objects. This point distribution, as expected from previous graphs, is located in the range below the 30-meter distance from the LiDAR sensor, represented by the red box. For the IoU metric, higher point accumulation occurs for an error under 0.25 and a distance less than 30 m, representing 55.43% of all objects.

##### Convolution Layer Hardware Replacement

As described in work [6], the backbone stage is built with sixteen convolution layers through each of the three blocks, which requires a lot of DSP and memory hardware resources, only two layers of block 3 were hardware processed. The inference process was done using a robotic operating system (ROS) platform that runs a new point cloud frame while initiating the PointPillars network for object detection and classification.

Table 2 and Table 3 present the obtained results for each evaluated metric regarding point cloud object detection. The score metric represents the probability of an object belonging to a certain class, in this example, cars. The location metric, expressed in meters, provides a spatial object identification on the point cloud. Each object position is given regarding the LiDAR sensor. The bounding box metric represents the BBOX around an object. Rotation_y and Alpha are related to the observation angle for each object.

From the obtained results, it is possible to notice there is not a huge divergence between the SW and hybrid versions, which uses values from hardware processing. During the hybrid model inference process it was verified that the number of false positives increases for the same score threshold value used during SW-only inference. To filter some of the false positives, which do not provide any valuable information about object detection or classification, the threshold value was increased from 0.5 to 0.75. However, the hybrid model presents one detection loss regarding the furthest car (car 6), as Figure 18 depicts and for the lack of metric results in Table 3. This detection loss was expected as expressed from a previous study exploited in Figure 16, which provides a visual perception that further objects tend to be less detected and present lower accuracy.

## 6. Conclusions

This paper had as its main goal the design and implementation of a convolution block, with the particularity of being totally customizable and applicable to any 3D object detection model. To increase parallelism for each convolution operation, the possibility of having several processing elements operating at the same time was added to the proposed architecture to improve inference time at the cost of energy and consumed logical resources.

Along with the development and implementation of the proposed generic convolution block, a study was conducted regarding the influence of quantization and parameter sharing. The quantization process, which reduces the bit-width of each parameter value, enables the second possible optimization that is related to “parameter sharing”. The bit-width reduction of weight values promotes a decrease of DSP usage of around 40% to 50%.

The developed IP was validated using RGB and point cloud data, in each evaluation it was verified that the proposed solution was capable of adaption regarding different model requirements. In the case of the model using point cloud data, with the PointPillars model it was possible to verify that higher scores and IoU values tend to appear near the LiDAR sensor, around less than 30 m. Furthermore, the quantization process affected both score and IoU values up a 10% decrease. Using the developed IP we visually confirmed the correct operation of the proposed solution. It was also possible to qualitatively validate the correct operation of the integration setup, the model being able to detect objects within a range of 30 m from the LiDAR.

## Figures and Tables

**Figure 1 sensors-22-02184-f001:**
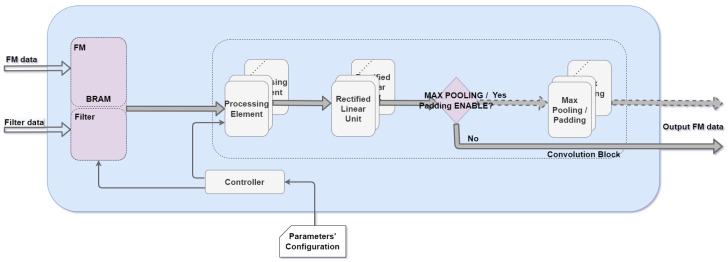
Convolution block architecture overview.

**Figure 2 sensors-22-02184-f002:**
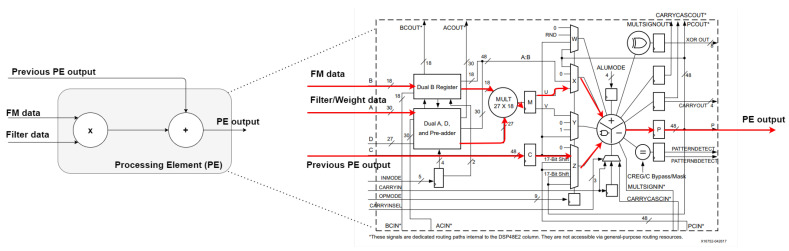
Processing element simplified block diagram on the left side. The right side illustrates a detailed version of DSP48E1 block diagram with input data path highlighted [31].

**Figure 3 sensors-22-02184-f003:**
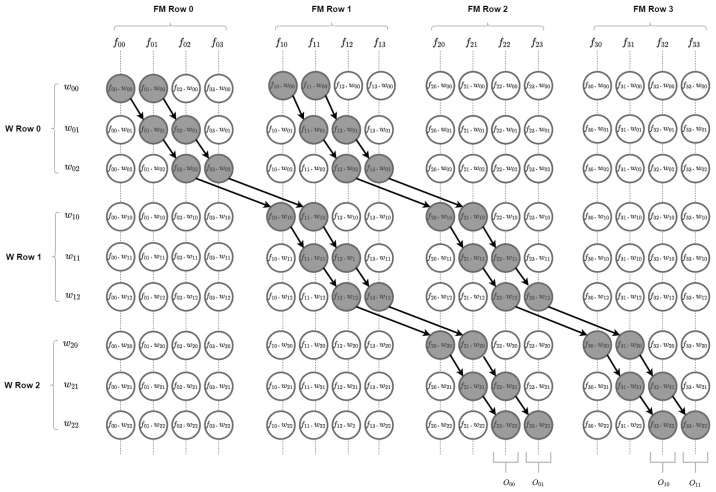
Dependency dataflow for a 4 × 4 FM, a 3 × 3 filter with stride = 1 and padding = 0.

**Figure 4 sensors-22-02184-f004:**
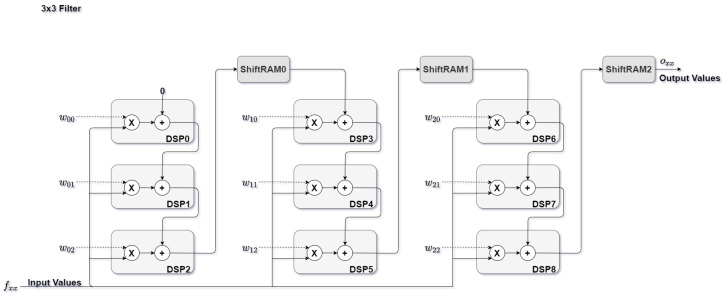
Processing element configuration for a 3 × 3 filter.

**Figure 5 sensors-22-02184-f005:**
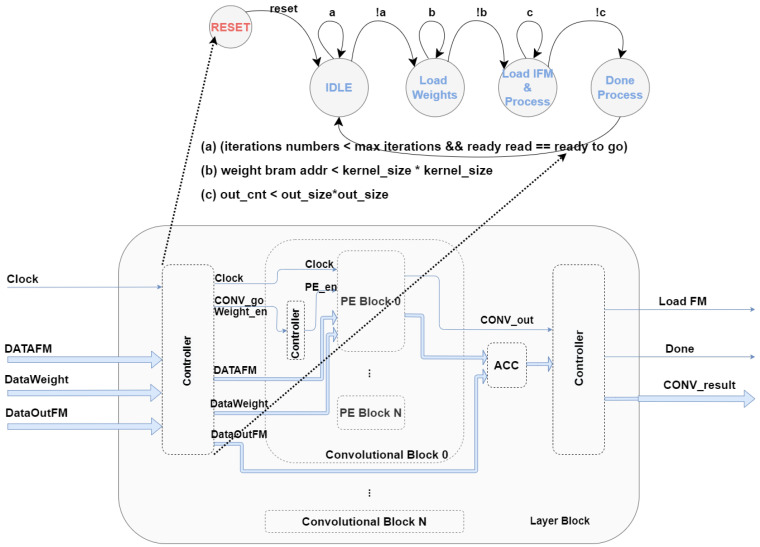
State machine diagram with stages interaction.

**Figure 6 sensors-22-02184-f006:**
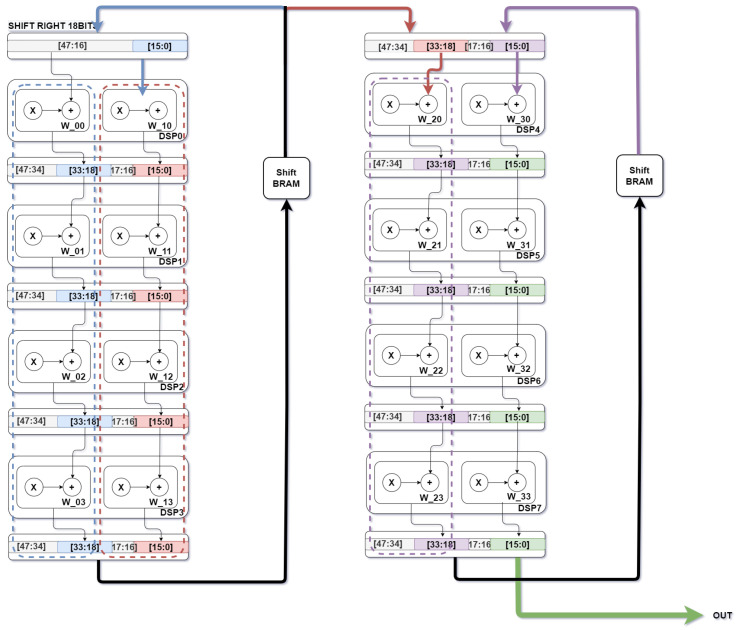
Block diagram for reconfigured architecture, providing two 8 bit products per DSP block.

**Figure 7 sensors-22-02184-f007:**
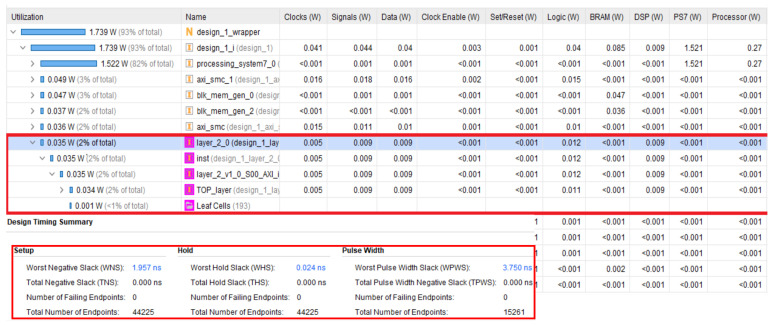
Resources usage and timing analysis for an image of 252 × 252 with a filter of 3 × 3, using only 1 PE.

**Figure 8 sensors-22-02184-f008:**
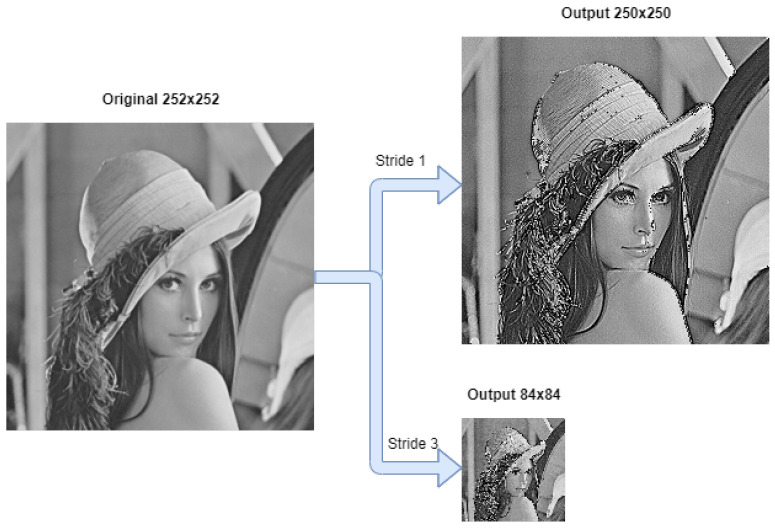
Right upper output image represents a sharpen filter application with stride 1, padding 0. Right bottom image represents a sharpen filter application with stride 3, padding 0.

**Figure 9 sensors-22-02184-f009:**
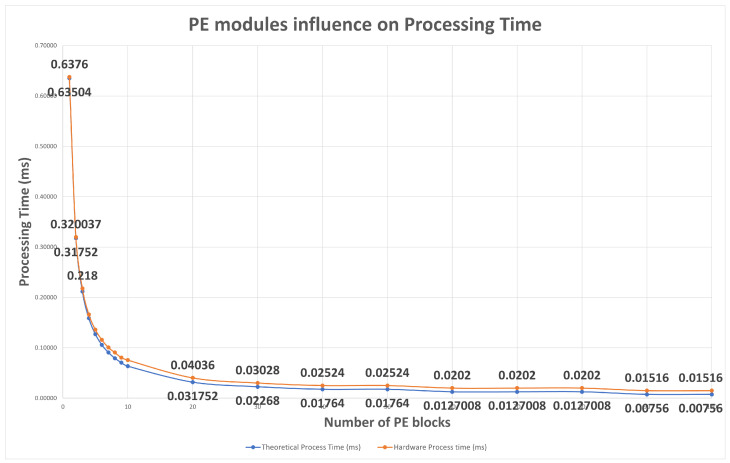
Study of PE modules influence on processing time. It begins with 1 PE block and finishes with 100 PE blocks.

**Figure 10 sensors-22-02184-f010:**
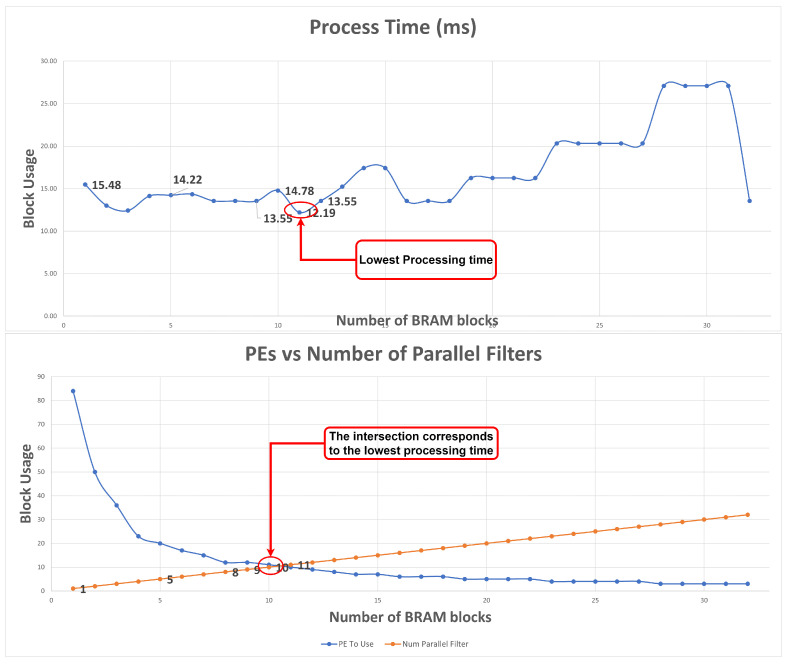
Number of PEs to use, number of parallel filters and processing time due to BRAM data.

**Figure 11 sensors-22-02184-f011:**
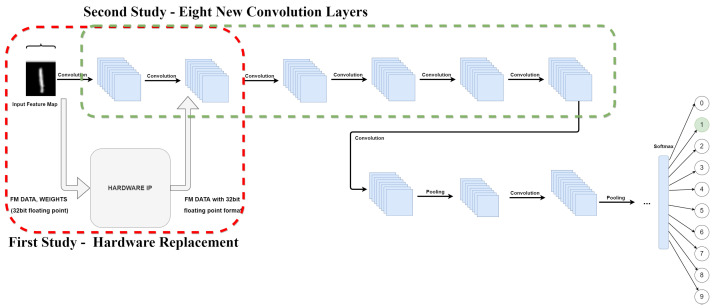
Two quantization studies on the MNIST model. Hardware replacement and influence on model performance as a function of model depth.

**Figure 12 sensors-22-02184-f012:**
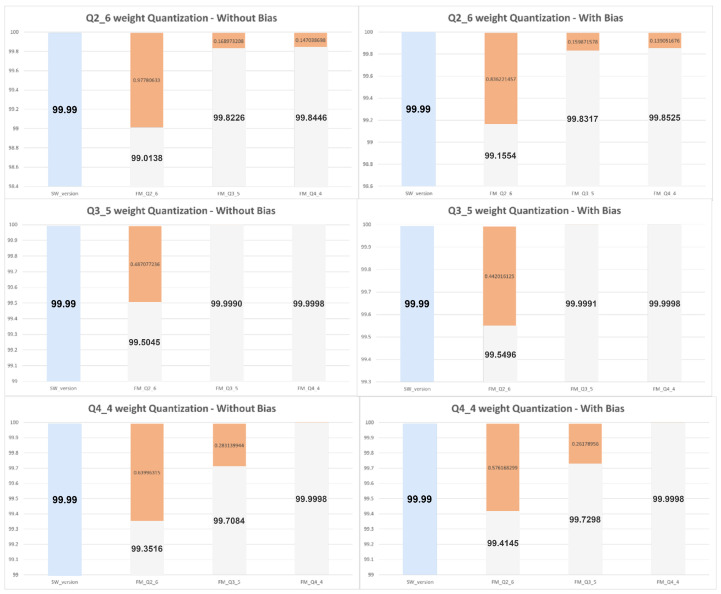
Evaluate classification score for Q2_6/Q3_5/Q4_4 weight quantization and Q2_6, Q3_5, and Q4_4 FM quantization. Blue bar provides SW data, grey quantized data and orange bar provides difference from SW_version and each quantized format.

**Figure 13 sensors-22-02184-f013:**
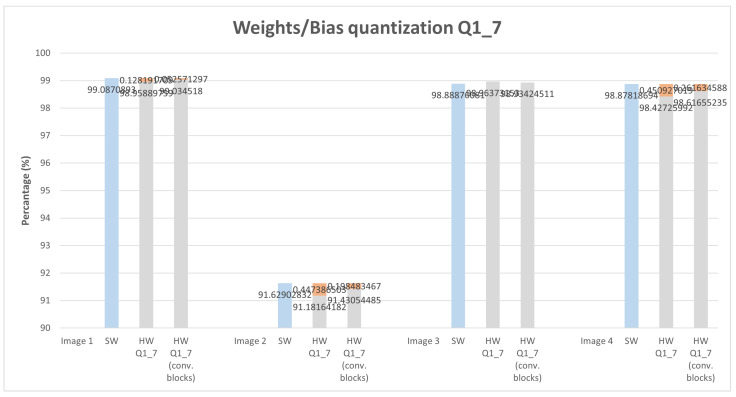
Weight and bias quantization for all MNIST model layers, and only for convolution layers with Q1_7. Blue bar provides SW data, grey quantized data and orange bar provides difference from SW_version and each quantized format.

**Figure 14 sensors-22-02184-f014:**
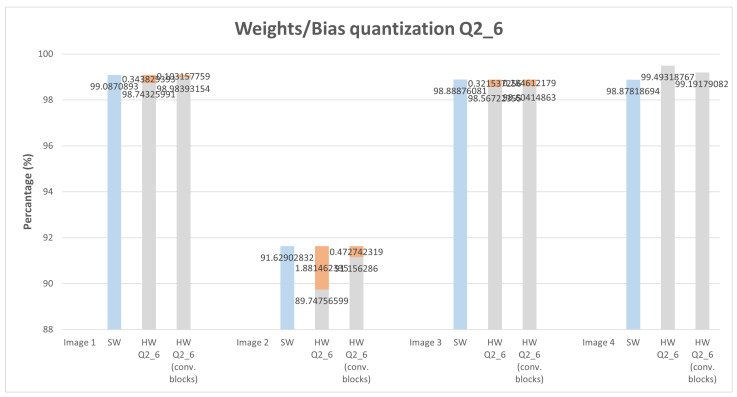
Weight and bias quantization for all MNIST model layers, and only for convolution layers with Q2_6. Orange bar provides difference from SW_version and each hardware quantized format.

**Figure 15 sensors-22-02184-f015:**
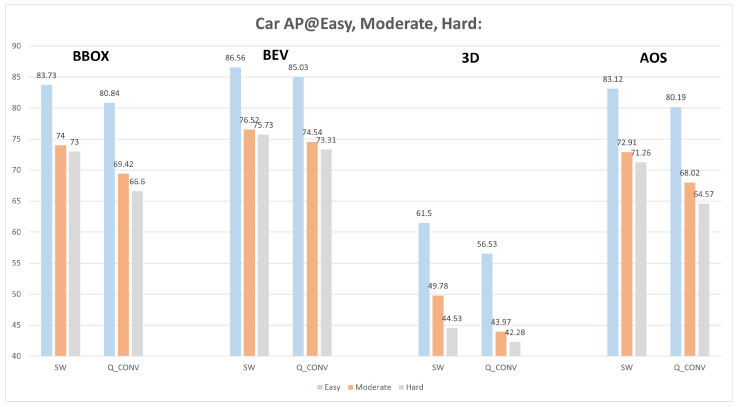
Evaluation between SW-only version and quantized model, with BBOX, BEV, 3D, and AOS metrics for easy, moderate, and hard difficulty levels.

**Figure 16 sensors-22-02184-f016:**
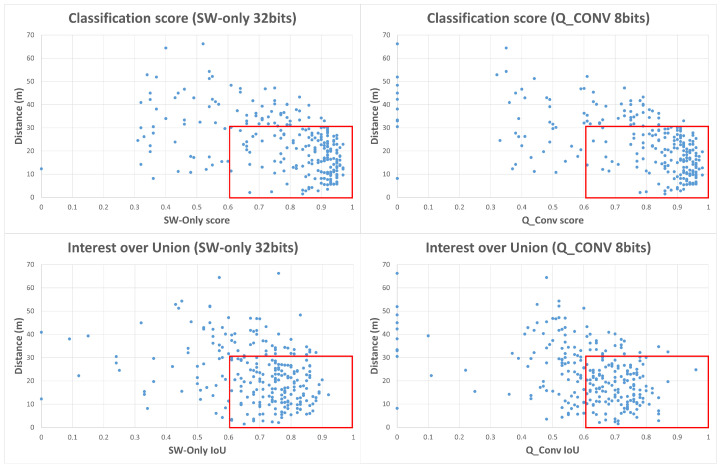
Scoreclassification and IoU distribution regarding object distance to LiDAR sensor. Left presents score and IoU for SW-only model version, right side presents results for quantized backbone convolution weights.

**Figure 17 sensors-22-02184-f017:**
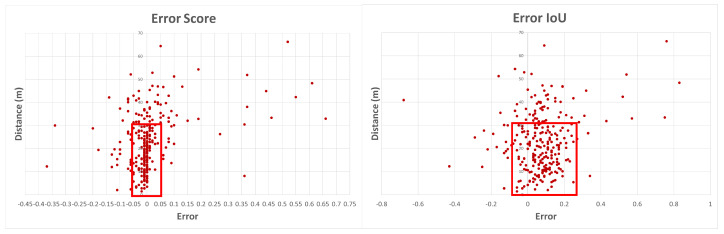
Error between SW-only model version and quantized backbone convolution weights, classification score, and IoU metrics.

**Figure 18 sensors-22-02184-f018:**
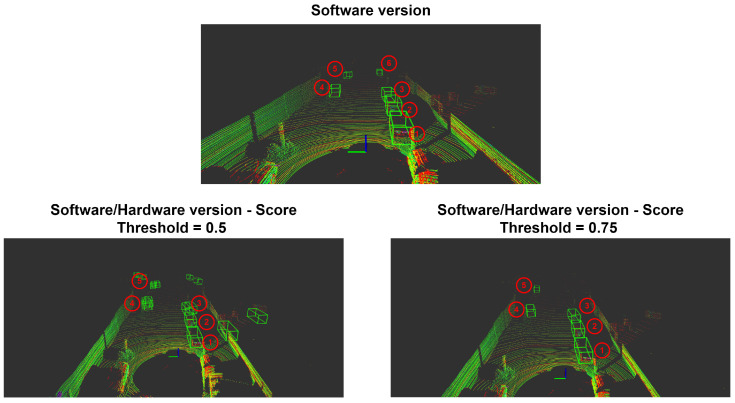
Point cloud frame displaying six cars with bounding box representation labeled from 1–6. SW–HW (hybrid) output detection for different score threshold value, left side with 0.5 and right side with 0.75.

**Table 1 sensors-22-02184-t001:** Resources used for convolution IP and all integrated design (central direct memory access (CDMA) module, blocks RAM and Zynq PS).

Clock Source	Convolution of a 252 × 252 FM with 3 × 3 Filter			
100 MHz	LUT	FF	DSP	BRAM
Convolution IP	2044	993	9	0
All Design	10,832	11,425	9	32.5 blocks

**Table 2 sensors-22-02184-t002:** Software only model version metrics for the evaluated frame.

SW-Only	Score (0…1)	Location x, y, z (m)	BBOX (Left, Top, Right, Bottom)	Rotation_y	Alpha
Car 1	0.90	2.88, 1.74, 6.42	770.46, 201.58, 1219.04, 370	4.68	4.27
Car 2	0.95	2.96, 1.54, 13.24	702.26, 182.34, 843.80, 277.94	4.69	4.47
Car 3	0.86	2.85, 1.55, 19.36	677.58, 181.34, 745.63, 243.11	4.78	4.63
Car 4	0.84	−6.13, 1.88, 23.85	373.33, 191.57, 463.90, 241.74	1.67	1.92
Car 5	0.72	−6.55, 1.69, 46.78	489.81, 182.45, 518.89, 207.38	1.49	1.62
Car 6	0.73	2.70, 1.03, 50.08	630.30, 173.84, 655.06, 195.71	4.71	4.66

**Table 3 sensors-22-02184-t003:** Software–hardware model version metrics for the evaluated frame.

SW-HW	Score (0…1)	Location x, y, z (m)	BBOX (Left, Top, Right, Bottom)	Rotation_y	Alpha
Car 1	0.77	2.52, 1.51, 6.55	753.33, 188.01, 1102.09, 370	4.74	4.39
Car 2	0.83	2.55, 1.39, 12.97	691.09, 177.49, 813.00, 269.46	4.74	4.54
Car 3	0.83	2.58, 1.31, 19.16	670.47, 175.11, 735.29, 234.51	4.79	4.66
Car 4	0.85	−5.81, 1.69, 23.56	379.53, 186.77, 472.51, 236.24	4.89	5.13
Car 5	0.82	−6.45, 1.52, 46.16	487.63, 180.95, 521.33, 204.80	4.74	4.88
Car 6	-	-	-	-	-

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
