# Peer review of "Customizable FPGA-Based Hardware Accelerator for Standard Convolution Processes Empowered with Quantization Applied to LiDAR Data"

_sensors, 2022, doi:10.3390/s22062184_

Round 1

Reviewer 1 Report

1) One of the major issues is the authors didn't discuss how many bits are the processed data and weights. The reviewer assumes it is double precision.   

2) Normally quantization should be done first, then use single-precision or half-precision if the impact on accuracy is neglectable. 
3) The experiment should be revised. Add the suggested analysis results to the manuscript. 

Figures: 

4) Fig 1, 2, 3, 4, 5, 6, 10, 11, 12, 13, 14, 15, 16, 17, 18 are barely readable. Increase font size for letters and numbers. 

5) Fig 12 Line 371 purple is hard to see. Use a different color 

6) Fig 7 is hard to read and unnecessary if the authors want to save on space. Remove Fig 7 and add a text description of the IP configuration to the main content. 

7) Fig 8 Timing analysis is unreadable. Show the timing analysis in a large figure or multiple figures. Refer to DOI: 10.1109/ACCESS.2022.3141098 as an example. 

8) Table 3, the highlight is unnecessary. Remove. 

9) Fig 19, if the circled numbers are the car 1 to 6. Add sentences to the captions to describe what is shown in the figure.   

10) Extensive proofreading is needed. I list a few suggestions below: 

Line 32, aside from

46, an FPGA, 

46, the main goal centers on

55 an object

59 a 3D model

86 being

128 discarded missing ","

136 however missing ","

190 Art

197 be connected

Fig 5 title dup

292 an FPGA

297 and Table 1

301 are illustrated

398 an object

413 scores are

419 plotted

425 wrong preposition

460 wrong preposition

472 at a cost of

Author Response

We would like to thank you for the comments and suggestions for improvement of our article. Indeed, it will help to improve and clarify our research. We thank you very much for the suggestions given for our manuscript. We do believe that the suggested changes have improved the overall quality of the manuscript. Bellow, the reviewer can find the performed changes to the text and reply to the questions raised by the reviewer.

1) One of the major issues is the authors didn't discuss how many bits are the processed data and weights. The reviewer assumes it is double precision.
Many thanks, we agree with the reviewer. In 2.3 Optimization Methods section, we describe that most deep learning models use as a baseline a single-precision, which means 32bit floating point for feature map and weight data. Also, in this section, we address the desire to quantize those 32bit floating point values to 16/8/4 bit fixed points. In fact, during 2.3 section, we did not explicitly refer to feature map or weight values in that 32bit floating point format. We have changed in section 3.6 to describe that data and weight values were quantized from 32bit floating point to 8bit fixed point.
2) Normally quantization should be done first, then use single-precision or half-precision if the impact on accuracy is neglectable. Many thanks for the suggestion. In this work, we had as baseline the software version of the MNIST model and PointPillars. Both of those works were developed using single precision, 32bit floating point, which provides the higher accuracy available during different analyses. Deploying a convolution block in hardware desires for a memory reduction to reduce chip area since fewer data is hardware deployed, for that reason, it was intended to quantize the weight values, which in this case was reduced from 32bit floating-point to 8bit fixed point. Since we already had a software version using single precision, it was not intended to evaluate a single-precision accuracy also on hardware, for that reason we exploit at first the quantization method.
3) The experiment should be revised. Add the suggested analysis results to the
manuscript. 
The other problems related to figures, tables and sentences expressions were solved

Reviewer 2 Report

The structure of this paper is reasonable, the discussion is rigorous, and the design of the experiment conforms to the specification, but there are still some small defects in some places. Here I offer some personal opinions:

When conducting the experimental analysis in Part 5, I think in addition to the comparison of the accuracy, the optimization degree of time should also be analyzed. After all, the main purpose of using FPGA is faster processing speed. When we use FPGA for design, we need to show how high the real-time improvement is when the result accuracy is not much different, so I suggest adding some graphs or tables showing the real-time improvement effect.

Some abbreviations are not explained, which will cause difficulty in understanding when reading, such as SW, HW, CDMA, etc., and some abbreviations are not necessary, such as Programmer Logic, because they are not used in the following text.

There are spelling or grammatical errors in some places, such as "too" on line 239, the name of Figure 5, the full name of BEV on line 410, and "Kittiviewer" on line 415.

There is a quoted display error on line 152, probably an encoding problem.

Author Response

We would like to thank you for the comments and suggestions for improvement of our article. Indeed, it will help to improve and clarify our research. We thank you very much for the suggestions given for our manuscript. We do believe that the suggested changes have improved the overall quality of the manuscript. Bellow, the reviewer can find the performed changes to the text and reply to the questions raised by the reviewer.

1) The structure of this paper is reasonable, the discussion is rigorous, and the design of the experiment conforms to the specification, but there are still some small defects in some places. Here I offer some personal opinions:
When conducting the experimental analysis in Part 5, I think in addition to the comparison of the accuracy, the optimization degree of time should also be analyzed. After all, the main purpose of using FPGA is faster processing speed. When we use FPGA for design, we need to show how high the real-time improvement is when the result accuracy is not much different, so I suggest adding some graphs or tables showing the real-time improvement effect.

Many thanks for the suggestion direction regarding processing time analysis. We analyze the optimization degree on 5.1.1 section where we evaluate the impact of PE modules, block RAMs, and the number of parallel filters on processing time. The input feature map size used in the quantization study is the same applied during processing time analysis. In fact, we do not provide a direct comparison between software and hardware version processing time. As with exploit in introduction section, with this work is not intended to create an entire CNN on an FPGA. The main goals centers on the implementation of a configurable convolution module, evaluation of the impact on performance by applying optimization methods such as quantization and parameter sharing, and integration of the convolution module on different CNN architectures. It allows to get a reliable perception of the impact of all approaches followed in the proposed article. This study is conducted in section 5.2. Even though there is not a direct comparison of a software version model and its hardware-based counterpart, we could determine the influence of the quantization in software before applying it on the hardware as analyzed in section 5.2.2 where we exploit all quantization to all feature map and weight data.

2) Some abbreviations are not explained, which will cause difficulty in understanding when reading, such as SW, HW, CDMA, etc., and some abbreviations are not necessary, such as Programmer Logic, because they are not used in the following text.

Many thanks for catching those problems. Those problems were solved by removing some unnecessary abbreviations and explaining some of the others.

3) There are spelling or grammatical errors in some places, such as "too" on line 239, the name of Figure 5, the full name of BEV on line 410, and "Kittiviewer" on line 415.

Those problems were solved in the new submission.

5) There is a quoted display error on line 152, probably an encoding problem.

Issue fixed in the new submisison.

Round 2

Reviewer 1 Report

Accept.